# White Matter Changes in Patients with Alzheimer’s Disease and Associated Factors

**DOI:** 10.3390/jcm8020167

**Published:** 2019-02-01

**Authors:** Yi-Hui Kao, Mei-Chuan Chou, Chun-Hung Chen, Yuan-Han Yang

**Affiliations:** 1Department of Neurology, National Taiwan University Hospital Yun-Lin Branch, Yun-Lin 640, Taiwan; blueggobi@gmail.com; 2Department of Neurology, Kaohsiung Municipal Ta-Tung Hospital, Kaohsiung City 801, Taiwan; neuro.chou@gmail.com; 3Department of Neurology, Kaohsiung Medical University Hospital, Kaohsiung Medical University, Kaohsiung City 807, Taiwan; mp245@ms49.hinet.net; 4Graduate Institute of Clinical Medicine, Kaohsiung Medical University, Kaohsiung City 807, Taiwan; 5Department of Neurology, Kaohsiung Municipal Siaogang Hospital, Kaohsiung City 812, Taiwan; 6Department of and Master’s Program in Neurology, Faculty of Medicine, Kaohsiung Medical University, Kaohsiung City 807, Taiwan; 7Neuroscience Research Center, Kaohsiung Medical University, Kaohsiung City 807, Taiwan

**Keywords:** Alzheimer’s Disease, age, diabetes mellitus, hypertension, Leukoaraiosis, white matter changes, vascular risk factor

## Abstract

Alzheimer’s disease (AD) is traditionally thought of as a neurodegenerative disease. Recent evidence shows that beta amyloid-independent vascular changes and beta amyloid-dependent neuronal dysfunction both equally influence the disease, leading to loss of structural and functional connectivity. White matter changes (WMCs) in the brain are commonly observed in dementia patients. The effect of vascular factors on WMCs and the relationship between WMCs and severity of AD in patients remain to be clarified. We recruited 501 clinically diagnosed probable AD patients with a series of comprehensive neuropsychological tests and brain imaging. The WMCs in cerebral CT or MRI were rated using both the modified Fazekas scale and the combined CT-MRI age related WMC (ARWMC) rating scale. Periventricular WMCs were observed in 79.4% of the patients and deep WMCs were also seen in 48.7% of the patients. WMC scores were significantly higher in the advanced dementia stage in periventricular WMCs (*p* = 0.001) and total ARWMCs (*p* < 0.001). Age and disease severity were both independently associated with WMCs score, particularly in the total, frontal and parieto-occipital areas. Vascular factors including hypertension, diabetes mellitus, and gender were not significantly associated with WMCs. In conclusion, both age and severity of dementia were significantly associated with WMCs in AD patients. These associations highlight future research targets.

## 1. Introduction

Alzheimer’s disease (AD) is widely accepted to be a disease of the cerebral gray matter. Nevertheless, some evidence has revealed the relationship of AD with both white matter changes (WMCs) and hippocampal atrophy [1]. Several pathological changes have been noted in the white matter, including the loss of axons, oligodendrocytes and reactive astrocytosis [2], dilation of the perivascular space, and failure of drainage of interstitial fluid secondary to deposition of beta-amyloid AD. Additionally, vascular insufficiency [3] and Wallerian degeneration of fiber tracts caused by neuronal loss in cortical associative areas [4] have also been related to the pathogenesis of WM changes in AD.

White matter changes (WMCs) have been reported to be associated with a decline in motor function in speed and fine motor coordination, and with many diseases including AD [2,5,6], vascular dementia, dementia with Lewy bodies, and psychiatric disorders [5]. However, WMCs were once considered to be benign and not related to dementia [7] as they are frequently present in the brain images of healthy aging.

Recent evidence show that both beta amyloid-independent vascular changes and beta amyloid-dependent neuronal dysfunction equally influence disease, leading to loss of structural and functional connectivity [8,9]. The two-hit vascular hypothesis [8,10] suggests that vascular factors, especially hypertension and DM play a role in vascular damage, which causes neurovascular dysfunction. Hypertension, especially in midlife, is a risk factor for AD [11]. The effect of vascular factors on WMCs and the relationship between WMCs and severity of AD in patients remain to be clarified. We conducted this study to investigate the characteristics of WMCs in AD patients and examine the association between vascular risk factors and WMCs.

## 2. Materials and Methods

### 2.1. Study Population

We retrospectively reviewed 501 patients with probable AD from the outpatient clinic of Kaohsiung Municipal Ta-Tung hospital, and who were diagnosed between October 2010 and April 2014. All patients received a complete evaluation for dementia, including demographic data, past medical history, physical and neurological examinations, comprehensive laboratory tests, and neuropsychological tests. Every patient underwent non-contrast brain CT or cerebral MRI scans upon the diagnosis of AD. Neuropsychological tests, including the clinical dementia rating [12] (CDR), Mini Mental State Examination (MMSE) and Neuropsychiatric Inventory (NPI) were performed by the clinical psychologists. The diagnosis of Alzheimer’s disease was made using the DSM-IV criteria and determined by experienced neurologists based on clinical symptoms, brain imaging, laboratory examinations of blood samples, and reference to neuropsychological assessments. Clinical Dementia Rating Scale (CDR) is a severity rating range along a 5-point scale (except for the personal care domain).

CDR-0: no cognitive impairment

CDR-0.5: questionable or very mild dementia

CDR-1: mild

CDR-2: moderate

CDR-3: severe

CDR = 0.5 is also commonly named mild cognitive impairment (MCI). MCI is considered to be a transitional stage between normal aging and AD [13]. The annual rate of transformation from questionable to mild dementia (CDR = 0.5 to 1) is around 10~15% [14]. We included the most important 3 intermediate stages of AD (0.5, 1 and 2) and took MCI as our benchmark.

The diagnoses of hypertension (HTN) and diabetes mellitus (DM) were based on patient-reported medical histories assisted by their medical records. Patients with previous stroke history were excluded. Moreover, all patients were strictly authorized to use acetylcholinesterase inhibitor (ACEI) therapy by independent neurologists from the National Health Insurance Administration, Ministry of Health and Welfare, Taiwan. This study was conducted in accordance with the Helsinki Declaration and was approved by the Institutional Review Board of Kaohsiung Medical University Hospital (KMUHIRB-20140063).

### 2.2. Evaluation of White Matter Changes

Cerebral CT or magnetic resonance imaging (MRI) scans with standard protocols were performed on every patient. One experienced neurologist, who was blind to all clinical information, rated the presence, location and severity of WMCs. WMCs were rated using both the modified Fazekas scale [6] and the combined CT-MRI visual age related WMC (ARWMC) rating scale [15]. In the Fazekas scale, the degree of white matter changes is rated on a 4-point scale as periventricular WMCs (PVWMCs) and deep WMCs (DWMCs). The Fazekas scale [6] was first proposed in 1987 to simply quantify the amount of white matter T2 hyperintense lesions. It is the most frequently used scale for white matter changes because of its simplicity and applicability on CT and MRI [16].

The CT-MRI visual ARWMC scale is a new rating scale with proven, almost equal sensitivity and applicability to both CT and MRI [15]. It is a modified version of the Fazekas scale and expands analysis in topographical regions. There are no differences between modalities in the frontal area and basal ganglia, however, in the parieto-occipital and infratentorial areas, MRI detects more WMC than CT does. Both CT and MRI can detect equally well in larger lesions. In order to sample large-scale samples, we used a rating scale established for both MRI and CT. In the CT-MRI visual ARWMC scale, the degree of white matter change was also rated on a 4-point scale, including five different regions in both the right and left hemispheres. We then summed the right and left sides according to each brain region including the total ARWMCs, frontal ARWMCs, parieto-occipital ARWMCs (P-O ARWMCs), temporal ARWMCs, infratentorial ARWMCs and basal ganglia ARWMCs [15]. In total, 463 CT images and 38 MRI images were rated. The neurologist responsible for rating the images was highly reliable in their ability to determine PVWMCs and DWMCs based on previous collaboration with researchers in Taiwan [17] and in the Faculty of Medicine at the Chinese University of Hong Kong (intraclass correlation coefficient = 0.810, 95% confidence interval: 0.601 to 0.910, *p* < 0.001).

### 2.3. Statistics

Statistical analysis was performed using SPSS Statistics version 20 (IBM, Armonk, NY, USA). All statistical tests were 2-tailed, and an alpha of 0.05 was considered to be significant. An analysis of variance (ANOVA) and Bonferroni’s post-hoc analysis were used when comparing WMC scores across different disease severity groups. A chi-squared test was used when comparing categorical variables in our analysis of the association of WMCs and factors including gender, CDR, hypertension (HTN) and diabetes mellitus (DM). To evaluate age, gender and risk factors to WMCs, we performed a binary logistic regression analysis using the modified Fazekas scale [6] and combined CT-MRI visual ARWMC rating scale [15] as the dependent variables.

## 3. Results

In this study, we included 501 patients with very mild to moderate AD; Table 1 shows the demographic characteristics of the patients. The mean age of the patients was 77.9 ± 7.7 years (range: 56–95); 69.3% of the patients were women. Among all of the patients, 249 (49.7%) had HTN and 120 (24%) had DM. The average duration of their education was 6.6 ± 5.2 years, and the average MMSE was 16.0 ± 6.2. The distribution of dementia severity was 128 (24.5%) for very mild dementia, 283 (56.5%) for mild dementia and 90 (18.0%) for moderate dementia.

Of our patients, 79.4% had PVWMCs and 48.7% had DWMCs. Table 2 shows the distribution of the Fazekas scale at different dementia stages. Even in the very mild dementia group, 72.7% had PVWMCs and 41.4% had DWMCs. Among the most severe group, as many as 84.4% had PVWMCs and 54.4% had DWMCs. Besides, PVWMCs are around 60% more frequently seen than DWMCs in these 3 groups. Characteristic images of WMCs are shown in Figure 1.

Table 3 shows that the WMC scores were significantly different across different disease severities for PVWMCs (*p* = 0.001), total ARWMCs (*p* < 0.001, frontal ARWMCs (*p* < 0.001) and parieto-occipital ARWMCs (P-O ARWMCs) (*p* < 0.000). The mean WMC scores increased with dementia severity for all ratings. Post hoc analysis showed that the CDR = 2 group had significantly higher scores than both the CDR = 0.5 and CDR = 1 groups.

The associations between WMCs and risk factors are shown in Table 4. CDR is the strongest factor influencing white matter scores, including PVWMCs (*p* = 0.012), total ARWMCs (*p* < 0.001), frontal ARWMCs (*p* < 0.001) and P-O ARWMCs (*p* < 0.001). Moreover, gender was related to PVWMCs (*p* = 0.028) and frontal ARWMCs (*p* = 0.014). HTN showed an association with PVWMCs (*p* = 0.030) and frontal ARWMCs (*p* = 0.028). DM did not correlate with any WMC scores.

Furthermore, we adjusted all results for gender, age, CDR, HTN and diabetes using a logistic regression model. As shown in Table 5, age was independently associated with PVWMCs (*p* < 0.0001), DWMCs (*p* = 0.019), total ARWMCs (*p* < 0.001), frontal ARWMCs (*p* < 0.001), P-O ARWMCs (*p* < 0.001) and basal ganglia ARWMCs (*p* < 0.001). CDR showed a significant correlation to total ARWMC score (*p* = 0.007), frontal ARWMC score (*p* = 0.004) and P-O ARWMC scores (*p* = 0.006). Following adjustment, gender, HTN and DM did not achieve significance.

## 4. Discussion

Based on our results, both periventricular and deep WMCs are common in Taiwanese AD patients. Our study indicates that the frontal areas have the most WMCs followed by the parieto-occipital areas, which is similar to reports in western AD patients [18]. The spatial distribution of WMCs is a reasonable consequence of the normal pattern of compromised perfusion in the frontal and posterior horns of the lateral ventricles found in AD and healthy aging [19]. Pathology reports show that sclerotic changes in the arteries supplying white matter are most prominent in the frontal lobe, followed by the parietal, occipital, and temporal lobes [20]. Moreover, WMCs in the frontal and parieto-occipital regions are particularly associated with hippocampal atrophy [1], which is a typical imaging finding in Artzokis et al. have demonstrated that AD is associated with a greater severity in myelin breakdown in the frontal lobe white matter [21]. Therefore, WMCs may be a secondary phenomenon of axonal change due to cortical neuronal damage in AD patients. Lastly, there is a large area of white matter in the frontal and parietal lobes, and an over-representation of those regions in axial slices [22] was further magnified by the visual rating scale. Even in patients with MCI, the cerebral blood flow was significantly reduced in the parietotemporal regions compared with healthy control [23]. This evidence supports the findings that WMCs in the frontal lobe followed by the parieto-occipital lobes is a common presentation in AD patients.

Our results show that age is strongly associated with multiple parameters, including PVWMCs, DWMCs, total ARWMCs, frontal ARWMCs, P-O ARWMCs and basal ganglia ARWMCs. It has also been reported that age is the primary indicator for cerebral WMCs in the normal elderly in addition to patients with AD [24] and stroke [17]. A probable mechanism for this is the age-related decrease in total cerebral blood flow that can be aggravated in aging brain pathologies [25], including tortuous arteries and periventricular venous collagenosis. The particular blood supply for the cerebral white matter lacks anastomoses [26] between the subependymal arteries and penetrating arteries from the cortical surface, making the white matter vulnerable to the aging microvascular environment and resulting in the widely observed white matter changes. Therefore, we believe that the universal effect of age could explain the variation in WMC prevalence across different studies.

In some small scale studies that included fewer than 50 AD patients [18,27], WMCs did not correlate with dementia severity, while others have reported that WMCs correlate with the degree of hypoperfusion [28] and cognitive impairment in the non-demented elderly. Our study shows that WMCs are significantly associated with CDR, especially total ARWMCs, frontal ARWMCs and P-O ARWMCs. PVWMC increases with CDR while DWMC does not. Moreover, moderate AD led to the accumulation of significant changes, especially in the frontal and P-O white matter and showed significantly higher scores. The most likely reason for this is that WMC is rated on a 4-point scale, which is so small that a large sample size is needed to show statistically significant differences between groups. Based on an abundant sample size, our findings show that the degree of WMCs correlated with the severity of AD. A greater number of WMCs superimposed on clinically diagnosed AD patients could have a synergistic effect and result in more severe symptoms. This concept is quite similar to that in the famous Nun study [29], which showed that brain infarcts were associated with poorer cognitive function in neuropathologically diagnosed AD. Recently, evidence has shown that an increased total WMC volume is an independent predictor for the diagnosis of AD [30] and that WMCs correlate with the degree of hippocampal atrophy [31].

In the Rotterdam study [32], women tended to have a higher degree of white matter change than men (total volume 1.45 mL v 1.29 mL; *p* = 0.33), particularly relative to frontal white matter changes (0.89 mL v 0.70 mL; *p* = 0.08). In another study, Sawada et al. showed that white matter was influenced by Alzheimer’s disease more in women than in men (*p* = 0.084). Similarly, we also found a trend for higher PVWMCs (*p* = 0.028) and frontal ARWMCs (*p* = 0.014) in women, but this was not statistically significant after adjustment. Female gender is known to be risk factor for AD, and a possible mechanism for this is that estrogens have antioxidant activities and neuroprotective effects [33]. Therefore, women of postmenopausal age could show an increased risk of AD.

Consistent with previous studies [20,34], HTN is associated with periventricular but not deep WMCs, and HTN primarily influences the frontal lobe. Chronic HTN debilitates cerebral autoregulation by damaging cerebral arterioles and capillaries [35], resulting in regional or diffuse cerebral hypoperfusion. Chronic hypoperfusion leads to an incomplete form of infarction and causes certain types of WMCs [34]. In demented patients, arteries that supply the white matter of the frontal lobe show the most sclerotic changes compared to other cerebral regions [20]. Therefore, the frontal lobe is the region most susceptible to hypertension, resulting in more WMCs. Moreover, in a 15-month follow-up study [36], HTN led to greater frontal deep WMC volume in healthy adults. The LADIS study also showed that HTN has effects on executive functions and attention, hinting at frontal lobe dysfunction.

The association between hypertension (HTN) and AD has been a subject of considerable debate. A previous study has shown that longitudinal follow up is key to understanding the association between HTN and AD [11]. Hypertension, especially midlife, is a risk factor for AD [11]. In our analysis, HTN tended to result in a higher score for PVWMCs (*p* = 0.030) and frontal ARWMCs (*p* = 0.028), although this effect was not statistically significant after adjustment. The lack of a longitudinal follow up could be the reason why hypertension seemed to be associated with periventricular WHC, but did not achieve significance in our study.

Some studies have shown that adult onset diabetes increases the risk of AD. According to Stewart [37], a possible mechanism for this association is the lack of a “mixed dementia” category such that all patients are dichotomously divided into AD or vascular dementia groups. Indeed, recent studies have demonstrated that diabetes is not associated with dementia or AD [38]. In our study, we strictly included AD without a vascular component and found that diabetes was not associated with any WMCs.

Two advantages of this study are the inclusion of a large sample size of AD patients and the fact that all of them were authorized to use acetylcholinesterase inhibitor (ACEI) by specialists from the National Health Insurance Administration, Ministry of Health and Welfare. With an abundant sample size, we were able to evaluate WMC locations and differences between severity groups. Currently, there are only a few studies that have focused on WMCs and AD in Asia [27]. Moreover, we have high inter-rater reliability for the rating of WMCs.

Our study has some limitations, such as the fact that that white matter changes are rated by a semi-quantitative scale with the inevitable disadvantages of a lack of sensitivity to small changes and a possible ceiling effect [6]. These disadvantages could be compensated for by the large sample size. Moreover, we included only very mild to moderate AD and did not include the duration and severity of risk factors in our retrospective design. Therefore, further study with longitudinal follow-up would be valuable for evaluating the relationship between WMC and disease progression.

In conclusion, both periventricular and deep WMCs are more common than expected in all stages of AD in patients. Age is the main indicator for cerebral WMCs in AD patients, and its universal effect causes the variation in prevalence across different studies. Our most remarkable finding is that the severity of the disease plays an important role in the WMCs, with significant differences between the moderate stage and milder stages of AD. Owing to their susceptibility to sclerotic changes in responsible arteries, myelin breakdown and over-representation on axial slices, the frontal lobe shows most WMCs followed by the parieto-occipital lobes. Females and patients with HTN tended to have a higher degree of WMCs in the periventricular and frontal areas, while there was no correlation with WMCs and diabetes. Further longitudinal follow up would be valuable and is necessary to clarify whether WMCs contribute to the cognitive decline in AD or they are just a sign of the disease process.

## Figures and Tables

**Figure 1 jcm-08-00167-f001:**
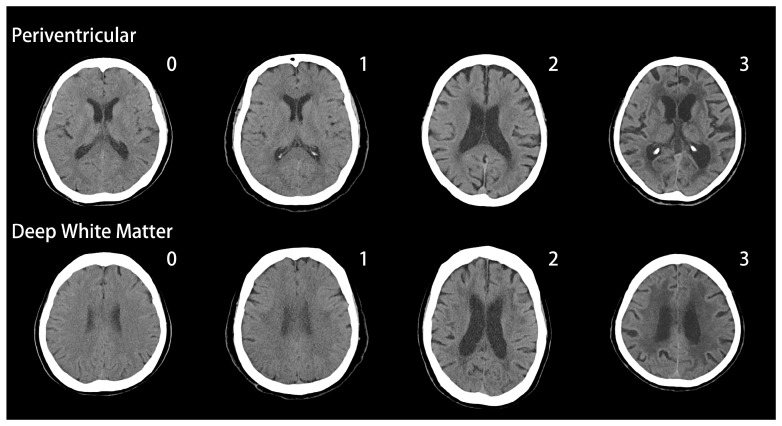
White matter changes (WMCs) in AD patients, Fazekas scale.

**Table 1 jcm-08-00167-t001:** Demographic characteristics of Alzheimer’s disease participants.

	N = 501
Age, year, mean (± SD)	77.9 (± 7.7)
Gender, female, *n* (%)	347 (69.3)
Hypertension, yes, *n* (%)	249 (49.7)
Diabetes mellitus, yes, *n* (%)	120 (24.0)
Education, year, mean (± SD)	6.6 (± 5.2)
MMSE *, mean (± SD)	16.0 (± 6.2)
CDR **	
CDR 0.5, *n* (%)	128 (24.5)
CDR 1, *n* (%)	283 (56.5)
CDR 2, *n* (%)	90 (18.0)

* MMSE = Mini-Metal Status Examination; ** CDR = Clinical Dementia Rating.

**Table 2 jcm-08-00167-t002:** Relationship between white matter changes and the severity of Alzheimer’s disease.

	AD			
Fazekas Scale		Very Mild Dementia CDR 0.5	Mild Dementia CDR 1	Moderate Dementia CDR 2
Periventricular white matter changes	0	35 (27.3%)	54 (19.1%)	14 (15.6%)
	1	44 (34.4%)	94 (33.2%)	18 (20.0%)
	2	22 (17.2%)	69 (24.4%)	23 (25.6%)
	3	27 (21.1%)	66 (23.3%)	35 (38.9%)
Deep white matter changes	0	75 (58.6%)	141 (49.8%)	41 (45.6%)
	1	19 (14.8%)	62 (21.9%)	15 (16.7%)
	2	18 (14.1%)	24 (8.5%)	6 (6.7%)
	3	16 (12.5%)	56 (19.8%)	28 (31.1%)

MMSE = Mini-Metal Status Examination; CDR = Clinical Dementia Rating; ARWMC = Age-Related White Matter Changes.

**Table 3 jcm-08-00167-t003:** Relationship between white matter changes and the severity of Alzheimer’s disease.

	AD N = 501				
	Total Patients	Very Mild Dementia CDR 0.5	Mild Dementia CDR 1	Moderate Dementia CDR 2	*p* Value
Fazekas scale (Periventricular)	1.5 ± 1.1	1.3 ± 1.1	1.5 ± 1.1	1.9 ± 1.1	0.001 *
Fazekas scale (Deep white matter)	1.0 ± 1.2	0.8 ± 1.1	1.0 ± 1.2	1.2 ± 1.3	0.067
ARWMC, total score	6.2 ± 5.8	5.1 ± 5.7	6.1 ± 5.6	8.2 ± 5.9	0.000 *
Frontal, score	2.6 ± 2.2	2.2 ± 2.2	2.6 ± 2.1	3.5 ± 2.2	0.000 *
Parieto-occipital, score	2.5 ± 2.3	2.1 ± 2.3	2.5 ± 2.2	3.3 ± 2.3	0.000 *
Temporal, score	0.8 ± 1.9	0.7 ± 1.8	0.8 ± 1.8	1.0 ± 2.1	0.515
Basal Ganglia, score	0.3 ± 0.7	0.2 ± 0.5	0.3 ± 0.7	0.4 ± 0.7	0.065
Infratentorial, score	0.0 ± 0.2	0.0 ± 0.1	0.0 ± 0.2	0.0 ± 0.3	0.359

MMSE = Mini-Metal Status Examination; CDR = Clinical Dementia Rating; ARWMC = Age-Related White Matter Changes. * Statistical significance was expressed by *p* value < 0.05.

**Table 4 jcm-08-00167-t004:** Factors related to white matter changes in Alzheimer’s disease patients.

	Gender (Female)	CDR	Hypertension	Diabetes Mellitus
Periventricular Fazekas scale	0.028 *	0.012 *	0.030 *	0.671
Deep white matter Fazekas scale	0.071	0.070	0.552	0.361
ARWMC, total score	0.091	0.000 *	0.083	0.234
Frontal, score	0.014 *	0.000 *	0.028 *	0.727
Parieto-occipital, score	0.168	0.000 *	0.259	0.191
Temporal, score	0.682	0.515	0.441	0.132
Basal Ganglia, score	0.494	0.065	0.103	0.685
Infratentorial, score	0.882	0.359	0.639	0.852

CDR = Clinical Dementia Rating; ARWMC = Age-Related White Matter Changes. * Statistical significance was expressed by *p* value < 0.05.

**Table 5 jcm-08-00167-t005:** Adjusted factors related to white matter changes in Alzheimer’s disease patients.

	Gender (Female) *p* Value (95% CI)	Age *p* Value (95% CI)	CDR *p* Value (95% CI)	Hypertension *p* Value (95% CI)	Diabetes Mellitus *p* Value (95% CI)
Periventricular Fazekas scale	0.110 (0.928–2.091)	0.000 *(1.048–1.106)	0.068 (0.973–2.140)	0.730 (0.722–1.591)	0.583 (0.722–1.785)
Deep white matter Fazekas scale	0.197 (0.860–2.082)	0.019 *(1.005–1.062)	0.536 (0.758–1.705)	0.891 (0.641–1.473)	0.175 (0.856–2.341)
ARWMC, total score	0.249 (−0.448–1.726)	0.000 *(0.065–0.199)	0.007 *(0.387–2.489)	0.148 (−1.837–0.277)	0.249 (−0.502–1.931)
Frontal, score	0.056 (−0.010–0.800)	0.000 *(0.042–0.093)	0.004 *(0.189–0.972)	0.167 (−0.671–0.116)	0.818 (−0.400–0.507)
Parieto-occipital, score	0.355 (−0.226–0.628)	0.000 *(0.030–0.083)	0.006 *(0.168–0.994)	0.406 (−0.591–0.240)	0.290 (−0.220–0.736)
Temporal, score	0.899 (−0.340–0.387)	0.732 (−0.026–0.019)	0.302 (−0.167–0.537)	0.201 (−0.584–0.123)	0.084 (−0.048–0.766)
Basal Ganglia, score	0.763 (−0.110–0.151)	0.004 *(0.004–0.020)	0.269 (−0.055–0.197)	0.176 (−0.214–0.039)	0.586 (−0.105–0.187)
Infratentorial, score	0.966 (−0.039–0.038)	0.698 (−0.002–0.003)	0.271 (−0.016–0.058)	0.630 (−0.047–0.028)	0.844 (−0.039–0.047)

CDR = Clinical Dementia Rating; ARWMC = Age-Related White Matter Changes. * Statistical significance was expressed by *p* value < 0.05.

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
