# Peer review of "White Matter Changes in Patients with Alzheimer’s Disease and Associated Factors"

_jcm, 2019, doi:10.3390/jcm8020167_

Reviewer 1 Report

While changes in glucose metabolism, gray matter atrophy, and amyloid deposition are most often investigated with MR or PET imaging, subtle white matter changes are thought to occur in the preclinical AD phase and in part account for cognitive decline. The authors sought to examine the effect of Alzheimer’s disease progression on white matter-related changes using well-established neurological rating scales, and if cardiovascular risk factors were also related to these changes.

For strengths, this is a larger sized study of an Asian population with varying degrees of AD and considerations of vascular factors. While various arms of ADNI have studied this question, in some cases with quantified WMH lesion load, all such studies overwhelming consist of Whites of European ancestry. The findings are comparable to previous studies, but the proportion of patients with type 2 diabetes was surprisingly high, as most literature suggests that Asian vs. White participants are less prone to developing cerebrovascular complications and by extension white matter damage.

For weaknesses, there are more sophisticated techniques for assessing WMHs. That being said, the resolution of CT and MRI are disparate, and if multiple units were utilized with varying resolution the semi-automated to automated techniques out there might be of limited use. I would also have liked to see a breakdown of whether MRI vs. CT was a significant nuisance variable, but this is a minor point.

Author Response

Dear reviewer:

I’m very grateful for your suggestions. I will describe more clearly here and also in the manuscript.

Fazekas scale[1] was first proposed in 1987 to simply quantify the amount of white matter T2 hyperintense lesions. It is most frequently used scale for white matter changes for its simplicity and applicability on CT and MRI[2].

A modified version of the Fazekas’ scale, the age related white matter changes (ARWMC) scale expanded analysis in topographical regions. According to the article “A New Rating Scale for Age-Related White Matter Changes Applicable to MRI and CT” published on Stroke[3], ARWMC scale is also a validated method used for evaluation of both CT and MRI that has almost equal sensitivity.

Despite unable to apply to automated techniques, visual assessment is still used worldwide and the most feasible hands-on method for numerous physicians including General Practitioner, Geriatrist and Internal Medicine Physician. This article would offer practical reference for most physicians taking care of AD patients.

Sincerely,

YIHUI KAO

Reviewer 2 Report

Authors studied CT and MRI of 501 clinical AD patients and investigated the WMCs with various affect factors. Authors find WMC score were significant higher in periventricular WMCs. The age and disease severity pf dementia were significantly associated with WMCs in AD patients. These kind association have been previous studied, and I didn’t see any novel point of this research. Furthermore, author didn’t include control group which is the health people. Patients received ether CT or MRI, in statistics, it is not convincible to combine data extracted from two different detection methods. Author didn’t provide any raw data like example of MRI images and showed where and what the changes looks like.

Author Response

Dear reviewer:

I’m grateful for the suggestions you mentioned. I will answer the points and describe clearly in the manuscript.

Due to the predicament of beta amyloid theory, the most recent concept about pathophysiology of AD is the two-hit hypothesis emphasis both on beta amyloid‑independent vascular changes and beta amyloid‑dependent neuronal dysfunction. Therefore, we reviewed previous articles about WMCs in AD patient and were surprised for the small case size. Fazekas[1] et al report only 12 Alzheimer patients and 4 patients with vascular dementia. Among them, there are 3 in the mild stage, 6 moderate stage and 3 severe AD. No any patient of CDR=0.5 was included. Other articles are in similar situation. The most important feature of our study is the sample size as compared to previous studies.

We design this study to show character of WMCs in AD patients and examine the associations between vascular risk factors and WMCs. Clinical Dementia Rating Scale (CDR) is a severity rating range along a 5-point scale (except for the personal care domain).

CDR-0: no cognitive impairment

CDR-0.5: questionable or very mild dementia

CDR-1: mild

CDR-2: moderate

CDR-3: severe

Among them, the CDR= 0.5 is also commonly named mild cognitive impairment (MCI). MCI is a considered to be a transitional stage between normal aging and AD[2]. The annual rate of transform from questionable to mild dementia (from CDR=0.5 to 1) is 10~15%[3]. We compared the most important 3 intermediate stages of AD including 0.5, 1 to 2 to present essential difference. Therefore, MCI itself is an adequate control group for study design. Besides, according to dynamic biomarkers of the Alzheimer’s pathological cascade[4], MCI just started to show subtle structure change on brain image. Compared with heterogeneous cognitively normal elderly, MCI is a suitable comparative target for demonstrate the transition.

Fazekas scale[1] was first proposed in 1987 to simply quantify the amount of white matter T2 hyperintense lesions. It is most frequently used scale for white matter changes for its simplicity and applicability on CT and MRI[5]. A modified version of the Fazekas’ scale, the age related white matter changes (ARWMC) scale expanded analysis in topographical regions. According to the article “A New Rating Scale for Age-Related White Matter Changes Applicable to MRI and CT” published on Stroke[6], ARWMC scale is also a validated method used for evaluation of both CT and MRI that has almost equal sensitivity. Despite unable to apply to automated techniques, visual assessment is still used worldwide and the most feasible hands-on method for numerous physicians including General Practitioner, Geriatrist and Internal Medicine Physician. Incorporated the latest concept, this article would offer practical reference for most physicians taking care of AD patients.

Besides, representative images of WM were shown in Fig1.

Figure 1. White matter changes (WMCs) in AD patients, Fazekas scale.

Sincerely,

YIHUI KAO

Reference:

1.         Fazekas, F.; Chawluk, J.B.; Alavi, A.; Hurtig, H.I.; Zimmerman, R.A., Mr signal abnormalities at 1.5 t in alzheimer's dementia and normal aging. American Journal of Neuroradiology 1987, 8, 421-426.

2.         Morris, J.C.; Storandt, M.; Miller, J.P.; McKeel, D.W.; Price, J.L.; Rubin, E.H.; Berg, L., Mild cognitive impairment represents early-stage alzheimer disease. Archives of neurology 2001, 58, 397-405.

3.         Boyle, P.; Wilson, R.; Aggarwal, N.; Tang, Y.; Bennett, D., Mild cognitive impairment risk of alzheimer disease and rate of cognitive decline. Neurology 2006, 67, 441-445.

4.         Jack Jr, C.R.; Knopman, D.S.; Jagust, W.J.; Shaw, L.M.; Aisen, P.S.; Weiner, M.W.; Petersen, R.C.; Trojanowski, J.Q., Hypothetical model of dynamic biomarkers of the alzheimer's pathological cascade. The Lancet Neurology 2010, 9, 119-128.

5.         Wahlund, L.-O.; Westman, E.; van Westen, D.; Wallin, A.; Shams, S.; Cavallin, L.; Larsson, E.-M., Imaging biomarkers of dementia: Recommended visual rating scales with teaching cases. Insights into imaging 2017, 8, 79-90.

6.         Wahlund, L.; Barkhof, F.; Fazekas, F.; Bronge, L.; Augustin, M.; Sjögren, M.; Wallin, A.; Ader, H.; Leys, D.; Pantoni, L., A new rating scale for age-related white matter changes applicable to mri and ct. Stroke 2001, 32, 1318-1322.

Reviewer 3 Report

This manuscript entitled “White matter changes in patients with Alzheimer’s disease  and associated factors” by  Kao et al. reports that the white matter changes (WMCs) score higher in Alzheimer’s disease (AD) patients with higher clinical dementia ratings (CDRs). While age was associated with WMCs, hypertension, diabetes mellitus and sex did not show association. The most prominent feature of this study is the sample size (501 patients) as compared to similar previously studies that were using ~50 subjects. As the authors speculate, small sample size seems to be the major cause for rather controversial conclusions those studies have reached. The authors performed thorough statistical analysis and discussed many current topics related to the results citing appropriate literature.

Major points

In addition to the lack of longitudinal follow-up study that the authors themselves admitted as disadvantage, the lack of normal control subjects is the major setback for this study. The logistic regression model used in Table 5 may contain overcompensation without the comparison with non-AD control subjects. This point should be fully discussed.

Major conclusion of this study is the association of WMC with CDR. To demonstrate the validity of WMC scoring, the authors should show representative images of WM of each CDR group (very mild, mild moderate).

Minor point

Line 179, the famous Nun study requires citation.     

Author Response

Dear reviewer:

I’m very grateful for your suggestions. I will answer the points and describe clearly in the manuscript.

As our title ” White Matter Changes in Patients with Alzheimer's Disease and associated factors”, we design this study to show character of WMCs in AD patients and examine the associations between factors and WMCs.

Clinical Dementia Rating Scale (CDR) is a severity rating range along a 5-point scale (except for the personal care domain).

CDR-0: no cognitive impairment

CDR-0.5: questionable or very mild dementia

CDR-1: mild

CDR-2: moderate

CDR-3: severe

Among them, the CDR= 0.5 is also commonly named mild cognitive impairment (MCI). MCI is a considered to be a transitional stage between normal aging and Alzheimer disease (AD)[1]. The annual rate of transform from questionable to mild dementia (CDR=0.5 to 1) is 10~15%[2]. We compared the most important 3 intermediate stages of AD including 0.5, 1 to 2 to present essential difference. Therefore, MCI itself is an adequate control group for study design. Besides, according to dynamic biomarkers of the Alzheimer’s pathological cascade[3], MCI just started to show subtle structure change on brain image. Compared with heterogeneous cognitively normal elderly, MCI is a suitable comparative target for demonstrate the transition.

Besides, representative images of WM were shown in Fig1.

Figure 1. White matter changes (WMCs) in AD patients, Fazekas scale.

Indeed, longitudinal follow-up study is our on-going work and we will report the results later. At last, I cited the famous Nun study. (Reference 29).

Due to the predicament of beta amyloid theory, the most recent concept about pathophysiology of AD is the two-hit hypothesis emphasis both on beta amyloid‑independent vascular changes and beta amyloid‑dependent neuronal dysfunction. We used to focus on gray matter and WMCs is often disregarded before. Therefore, we reviewed previous articles about WMCs in AD patient and were surprised for the small case size. Fazekas[4] et al report only 12 Alzheimer patients and 4 patients with vascular dementia. Among them, there are 3 in the mild stage, 6 moderate stage and 3 severe AD. No any patient of CDR=0.5 was included. Other articles are in similar situation. As you mentioned, the most important feature of our study is the sample size as compared to previous studies. Incorporated the latest concept, this article would offer practical reference for most physicians taking care of AD patients.

Sincerely,

YIHUI KAO

Reference:

1.         Morris, J.C.; Storandt, M.; Miller, J.P.; McKeel, D.W.; Price, J.L.; Rubin, E.H.; Berg, L., Mild cognitive impairment represents early-stage alzheimer disease. Archives of neurology 2001, 58, 397-405.

2.         Boyle, P.; Wilson, R.; Aggarwal, N.; Tang, Y.; Bennett, D., Mild cognitive impairment risk of alzheimer disease and rate of cognitive decline. Neurology 2006, 67, 441-445.

3.         Jack Jr, C.R.; Knopman, D.S.; Jagust, W.J.; Shaw, L.M.; Aisen, P.S.; Weiner, M.W.; Petersen, R.C.; Trojanowski, J.Q., Hypothetical model of dynamic biomarkers of the alzheimer's pathological cascade. The Lancet Neurology 2010, 9, 119-128.

4.         Fazekas, F.; Chawluk, J.B.; Alavi, A.; Hurtig, H.I.; Zimmerman, R.A., Mr signal abnormalities at 1.5 t in alzheimer's dementia and normal aging. American Journal of Neuroradiology 1987, 8, 421-426.

Round  2

Reviewer 2 Report

Thank you for your response. No more comments.